# A particle-based computational model to analyse remodelling of the red blood cell cytoskeleton during malaria infections

Julia Jäger[1,2], Pintu Patra[1,2], Cecilia P. Sanchez[3], Michael Lanzer[3]\*, Ulrich S. Schwarz[1,2]\*

**1** Institute for Theoretical Physics, Heidelberg University, Heidelberg, Germany, **2** BioQuant-Center for Quantitative Biology, Heidelberg University, Heidelberg, Germany, **3** Center of Infectious Diseases, Parasitology, University Hospital Heidelberg, Heidelberg, Germany

\* Michael.Lanzer@med.uni-heidelberg.de (ML); schwarz@thphys.uni-heidelberg.de (USS)

**Data Availability Statement:** Our custom-written scripts for ReaDDy 2 are available on GitHub as https://github.com/usschwarz/networks. All other

## Abstract

Red blood cells can withstand the harsh mechanical conditions in the vasculature only because the bending rigidity of their plasma membrane is complemented by the shear elasticity of the underlying spectrin-actin network. During an infection by the malaria parasite *Plasmodium falciparum*, the parasite mines host actin from the junctional complexes and establishes a system of adhesive knobs, whose main structural component is the knob-associated histidine rich protein (KAHRP) secreted by the parasite. Here we aim at a mechanistic understanding of this dramatic transformation process. We have developed a particle-based computational model for the cytoskeleton of red blood cells and simulated it with Brownian dynamics to predict the mechanical changes resulting from actin mining and KAHRP-clustering. Our simulations include the three-dimensional conformations of the semi-flexible spectrin chains, the capping of the actin protofilaments and several established binding sites for KAHRP. For the healthy red blood cell, we find that incorporation of actin protofilaments leads to two regimes in the shear response. Actin mining decreases the shear modulus, but knob formation increases it. We show that dynamical changes in KAHRP binding affinities can explain the experimentally observed relocalization of KAHRP from ankyrin to actin complexes and demonstrate good qualitative agreement with experiments by measuring pair cross-correlations both in the computer simulations and in super-resolution imaging experiments.

## Author summary

Malaria is one of the deadliest infectious diseases and its symptoms are related to the blood stage, when the parasite multiplies within red blood cells. In order to avoid clearance by the spleen, the parasite produces specific factors like the adhesion receptor PfEMP1 and the multifunctional protein KAHRP that lead to the formation of adhesive knobs on the surface of the red blood cells and thus increase residence time in the vasculature. We have developed a computational model for the parasite-induced remodelling of

relevant data are within the manuscript and its Supporting information files.

**Funding:** This work was funded by the Deutsche Forschungsgemeinschaft (DFG, German Research Foundation) through grants to ML and USS (Projektnummer 240245660 – SFB 1129). The funders had no role in study design, data collection and analysis, decision to publish, or preparation of the manuscript.

**Competing interests:** The authors have declared that no competing interests exist.

the actin-spectrin network to quantitatively predict the dynamical changes in the mechanical properties of the infected red blood cells and the spatial distribution of the different protein components of the membrane skeleton. Our simulations show that KAHRP can relocate to actin junctions due to dynamical changes in binding affinities, in good qualitative agreement with super-resolution imaging experiments. In the future, our simulation framework can be used to gain further mechanistic insight into the way malaria parasites attack the red blood cell cytoskeleton.

## Introduction

Malaria infections cause around 400.000 fatalities per year [1] and most of these are caused by the species *Plasmodium falciparum*. The main symptoms of this disease are related to the blood stage, when the malaria parasite hides inside a red blood cell (RBC) [2]. There it replicates to produce around 20 offspring that after 48 hours are released into the blood stream with the rupture of the RBC-envelope. During this development, the parasite removes most of the haemoglobin and causes a dramatic remodelling of the RBC-envelope [3]. Remodelling of the cytoskeleton is essential to prevent premature rupture and to establish a system of adhesive knobs. By cytoadhesion to the vascular endothelium, the residency time in the vasculature is increased and the parasite avoids clearance by the spleen. Strikingly, this dramatic transformation process is controlled by the parasite only from a distance, through secretion of proteins that are targeted to the membrane and there incorporated into the existing surface structures of the RBC [4]. However, the exact points of attack and the temporal sequence of these remodeling events are not yet understood.

The main component of the red blood cell cytoskeleton are spectrin tetramers which form a quasi-hexagonal network with in average six tetramers converging to the around 35.000 actin protofilaments, each of which is only 36 nm long [5, 6]. Spectrin tetramers consist of two head-to-head attached dimers and have a contour length of approximately 200 nm [6–8]. With around 10 nm, the measured persistence length is much shorter than this contour length [9–11]. Therefore the spectrin filaments are strongly coiled and generate a thick layer underneath the membrane [12–15]. Polymer theory predicts an end-to-end distance of 63 nm, which agrees well with the range of 50 to 100 nm that has been reported experimentally [5, 6]. However, the exact length distribution of the spectrin tetramers in equilibrium is not known, since all microscopy techniques require sample preparations that tend to stretch the spectrin chains.

The spectrin-actin cytoskeleton provides shear elasticity to the RBC-envelope and is closely connected to the plasma membrane, which provides bending rigidity [16]. Together, these two layers can withstand the harsh conditions that the RBC faces in the vasculature, ranging from the high shear rates in the heart to the strong confinement in the capillaries. In order for this composite to be mechanically stable, many connections are required between the two layers. These are established not only by the actin junctional complexes at the nodes of the hexagonal spectrin network, but also at the ankyrin complexes at the midpoints of the spectrin tetramers connecting these junctional complexes [5, 6] (Fig 1A). The two types of complexes are anchored to the membrane by actin binding to glycophorin C (GPC) and by ankyrin binding to anion exchanger 1 (AE1), respectively. AE1 is also known as band 3 and with 1.2 million copies is very abundant in RBCs because it localizes to both types of complexes and can be free in the membrane [6].

Long being considered to be relatively static structures, only recently it has been found that junctional complexes are in fact in a state of continuous turnover, as demonstrated

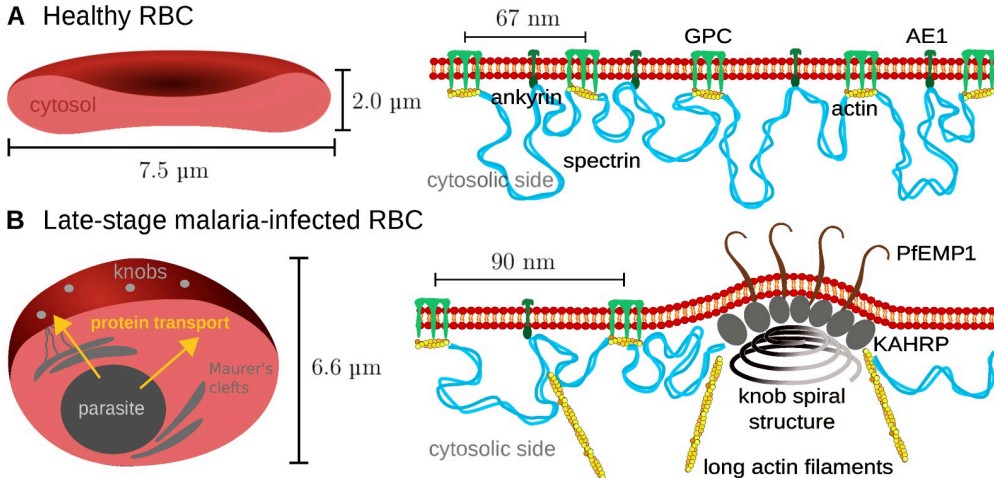

**Fig 1. The membrane skeleton in healthy and infected red blood cells. (A)** Left: schematic representation of a cut through a healthy RBC with a biconcave shape. Right: closeup of the coupling between the actin-spectrin network and the lipid bilayer. The actin protofilaments of the junctional complexes bind to glycophorin C (GPC) and ankyrin binds to anion exchanger 1 (AE1, also known as band 3). **(B)** Left: schematic representation of the late stage of the malaria infection, when the RBC has changed to a spherical shape and adhesive knobs have formed on its surface. Right: closeup of the internal structure of the knobs, with a spiral structure at the base, the PfEMP1 adhesion receptors in the membrane at the top and KAHRP-molecules inbetween. Actin is mined from the junctional complexes and forms long filaments often connected to the Maurer's clefts.

experimentally by actin monomer exchange and the discovery that not all actin filaments are capped [17–19]. Capping is provided by the two capping molecules adducin (for the fast-growing barbed end) and tropomodulin (for the slow-growing pointed end). In addition, tropomyosin stablizes the actin protofilament by binding at its side. The insight that the actin protofilaments are more dynamic than thought earlier was in fact originally triggered by the finding that malaria parasites mine actin from the junctional complexes, which is then used to build a new transport system of long filaments between the parasite and the RBC-surface (Fig 1B) [20, 21]. However, the exact molecular mechanisms underlying actin mining by the parasite are not known yet.

Changes in the spectrin-actin cytoskeleton have an immediate impact on the mechanics and quality control of RBCs. RBCs have a typical lifetime of 120 days and senescent but also malaria-infected RBCs (iRBCs) are removed from the circulation mainly at the interendothelial slits of the spleen. It has been found experimentally that the shear modulus of the membrane skeleton of iRBCs increases by one order of magnitude [22–24]. This stiffening but also the growing parasite mass inside make it very difficult to pass the slits. In order to avoid the clearance by the spleen, the malaria parasite has evolved cytoadhesion as a strategy to increase residence time in the vasculature. The parasite-produced *Plasmodium falciparum* erythrocyte membrane protein 1 (PfEMP1) is transported along the long actin filaments that result from the actin mining and are clustered in thousands of adhesion clusters called knobs (Fig 1B). These complexes are 100 nm large protrusions which can be imaged with electron microscopy or atomic force microscopy [25, 26]. Their main component is the knob-associated histidine rich protein (KAHRP) which is essential for knob formation [27–29]. Moreover it has been shown that a spiral scaffold of unknown composition underlies the knobs [30, 31]. Towards the tip of these knobs, on average three PfEMP1 molecules are clustered [32]. As the knobs are formed, also the spectrin network becomes more irregular, with a larger average length of spectrin filaments [33]. Coarse-grained molecular dynamics computer simulations have

suggested that the stiffening effect however results mainly from knob formation and less from the changes in the spectrin network [34].

Very recently, we have shown using two-color stimulated emission-depletion (STED) microscopy and image pair cross-correlation analysis that the KAHRP localization dynamcially changes during a malaria infection [35]. During most of the initial ring stage (1–24 hours post invasion), KAHRP co-localizes both with the ankyrin and actin junctional complexes, possibly to create a strong diffusion gradient towards the membrane. At the end of the ring stage and the beginning of the trophozoite stage (24–36 hours post invasion), it re-localizes to the remodelled actin junctional complexes, where most of the knobs seem to form. However, it is unclear which molecular processes are underlying these dynamical changes. Potential mechanisms are changes in KAHRP affinity due to phosphorylation, increases in KAHRP concentration and structural changes in the KAHRP target structures. To explore the spatial aspects of these dynamical changes on the molecular level and how they could explain the known mechanical changes on the cellular level, here we have developed coarse-grained Browninan dynamics simulations that now allow us to quantitatively analyse the dramatic remodelling process of RBCs that occurs during the infectious cycle of *Plasmodium falciparum*. Similar coarse-grained computer simulations have been used before to predict the cellular shear modulus from a microscopic description of the RBC cytoskeleton for healthy and infected RBCs [34, 36, 37], but not with the focus on the dynamical changes to actin and KAHRP as addressed here.

## Materials and methods

### Particle-based model for the cytoskeleton

The cytoskeletal network is simulated with ReaDDy 2 [38], which is a Brownian Dynamics simulation framework for interacting particles and the additional capabilities to incorporate reactions and filament assembly. Particles diffuse according to an overdamped Langevin equation with predefined potentials depending on nearby particle positions $V(\vec{x}_i(t), \vec{x}_j(t), \ldots)$:

$$\frac{d\vec{x}_i(t)}{dt} = -\frac{D_i}{k_B T} \nabla V(\vec{x}_i(t), \vec{x}_j(t), \ldots) + \sqrt{2D_i}\vec{\xi}_i(t), \tag{1}$$

where $\vec{x}_i(t)$ is the particle position at time $t$, $D_i$ is the particle-specific diffusion constant, $k_B T$ is thermal energy and $\vec{\xi}_i(t)$ is white noise:

$$\langle \vec{\xi}_i(t) \rangle = 0 \;, \quad \langle \vec{\xi}_i(t)\vec{\xi}_j^T(t') \rangle = \delta_{ij}\delta(t - t').$$

Cytoskeletal filaments and monomers are set up with specific interactions as explained hereafter. The simulation time step was chosen as $\Delta t = 0.01$ ns to account for the strong potentials within filaments. A typical simulation run lasts for 0.26 ms.

Filaments are implemented as a connected chain of monomers with harmonic bond potentials between neighbouring beads and angle potentials between three adjacent beads. The pairwise potentials are described in detail in S1 File. Through the angle coefficient $k_{angle}$ the persistence length $L_p$ of the filament is determined. For the cytoskeletal network of the RBC we need two different types of filaments, spectrin filaments of contour length of 205 nm and polymerizing actin filaments with an equilibrium length of 36 nm. We use $k_{angle} = 4.28$ kJ mol$^{-1}$ for the very flexible spectrin filaments (corresponding to a persistence length of 10 nm) and $k_{angle} = 4280$ kJ mol$^{-1}$ for the rod-like actin filaments (corresponding to a persistence length of 10 μm). All non-neighbouring particles are subject to a hard-core repulsion to model excluded volume effects. The force constants for the repulsion are given in S1 Table in S1 File.

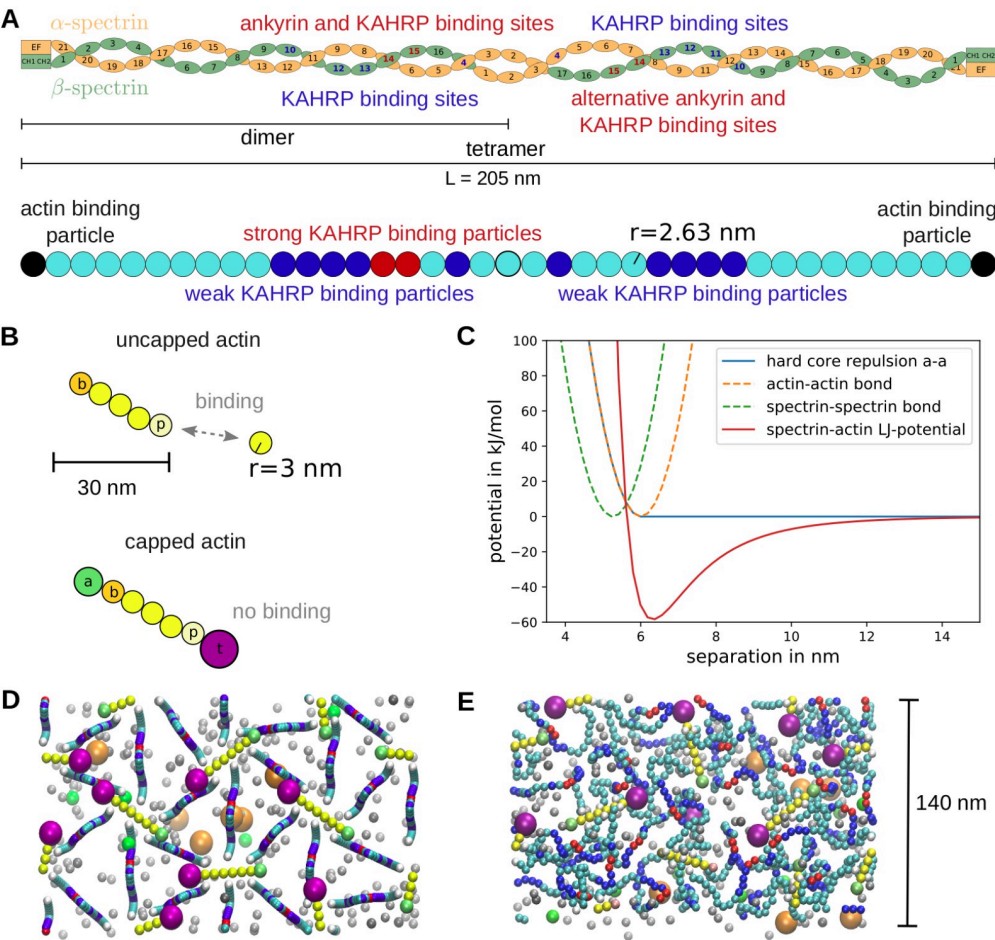

**Fig 2. Particle-based model for the RBC cytoskeleton. (A)** Schematic of the spectrin tetramer structure at the top and implementation in the simulation at the bottom. Differently coloured beads have different properties that represent the interactions of the spectrin filaments with actin, ankyrin and KAHRP. **(B)** Implementation of dynamic actin filaments in the simulations. The top filament can polymerize at both ends with different rates at the barbed (*b*) and pointed (*p*) ends. For the filament at the bottom the polymerization is blocked, since the capping proteins adducin (*a*) and tropomodulin (*t*) are attached. **(C)** The potentials used in the simulations are plotted against particle separation. **(D)** The initial configuration of a typical simulation is shown as a projection along the z-direction. The simulation box has periodic boundaries and a size of $140 \times 242.48 \times 100$ nm$^{-3}$. G-actin particles are shown in grey. An example simulation can be seen in the S1 Movie. **(E)** Equilibrated state of the network.

A spectrin dimer consists of 21 $\alpha$-spectrin repeats, 17 $\beta$-spectrin repeats and one actin binding domain at each end as shown schematically in Fig 2A top [6, 39]. During dimer formation, the $\alpha$- and $\beta$-chains of spectrin zip up from the actin-binding tail that is held together by strong electrostatic interactions. Next two dimers associate head-to-head to from a spectrin tetramer, which is believed to act as a single strand for most of the time, although the two chains might slide relatively to each other [6]. In the simulation we adopt the standard approach to model the spectrin tetramer as a worm-like chain (WLC) [37, 40]. Following earlier particle-based simulations of the spectrin network, we use 39 beads (Fig 2A bottom) [36, 37, 41]. With a bead radius of 2.63 nm as used earlier, this gives us a contour length of 205 nm. Because here we only simulate small patches of the skeleton, the membrane is not modeled explicitly, but only enters in its function to confine the two types of complexes to a similar height. In detail, we harmonically confine the ankyrin- and actin-binding spectrin beads to a 4 nm thick plane 10

 

nm above the bottom of the simulation box which represents the plane of the lipid bilayer. These dimensions are chosen as to represent the typical size of the transmembrane anchoring proteins. Ankyrin can bind to repeats 14 and 15 of the $\beta$-spectrin chain, so in principle there are two potential ankyrin binding regions on spectrin, namely left and right to the tetramerization site. However, only one ankyrin can bind at any time for steric reasons [6], thus in the simulations we randomly choose one of them for confinement (compare Fig 2A).

Actin filaments are implemented with two distinct ends corresponding to the barbed and pointed ends and can react with G-actin to elongate or shrink at predefined rates (compare Fig 2B). The filaments are confined to the 4 nm thick plane such that they maintain a low angle with respect to the bottom of the simulation box as has been observed in RBCs [14]. Within this plane the filaments are free to diffuse and rotate.

All particles that are confined to the membrane plane are set up with a reduced diffusion constant of 0.53 μm² s⁻¹ to represent their slowing down by the binding to transmembrane proteins like GPC and AE1. The reduced value of the diffusion constant is the one measured for AE1 diffusing on small scales within the RBC membrane [42]. For calculating the diffusion constant of the spectrin beads we took the doubled size of a single spectrin bead (see effective size in Table 1) in order to reflect the thickness due to two filament strands.

All monomers are assumed to be uniformly reactive and to have an isotropic diffusion constant. The diffusion constant was either taken from previous measurements or calculated from the Stokes-Einstein relation:

$$D_i = \frac{k_B T}{6 \pi \eta R_i},$$ (2)

with $\eta$ being the dynamic viscosity and $R_i$ the hydrodynamic radius of particle $i$. Values are summarised in Table 1. For the cytoskeletal simulations, three kinds of monomers are introduced in order to interact with actin filaments: G-actin, adducin and tropomodulin. G-actin can attach to the barbed or pointed ends and then becomes the new end bead. On the contrary, if one end shrinks, a G-actin monomer is released and the neighbouring particle becomes the end bead. Adducin and tropomodulin block this polymerization process by attaching to the barbed and pointed ends, respectively, and stop any further polymerisation until they detach again (compare Fig 2B). The effect of tropomyosin, which attaches laterally to the actin helix, is not modeled explicitly, but implicitly by enforcing the typical actin protofilament length (see below).

In order to model the effect of a malaria infection, we introduce KAHRP particles which can bind to cytoskeletal components and other KAHRP particles via Lennard-Jones potentials

**Table 1. Values for collision radii, reaction radii and diffusion constants.**

| particle type | $R_c$ in nm | $R_i$ in nm | $D$ in $\frac{\mu m^2}{s}$ |
|---|---|---|---|
| spectrin bead | 2.63 | 2.63 | not used |
| spectrin effective size | 5.26 | 5.26 | 40.78 |
| KAHRP | 2.8 | 2.8 | 76.6 |
| G-actin | 3 | 3 | 71.5 |
| adducin | 4.2 | 4.2 | 51.07 |
| tropomodulin | 7.25 | 7.25 | 29.59 |

The diffusion constant of G-actin is taken from Lanni *et al.* [43] and the other values are calculated from the respective molecular masses of the proteins assuming spherical particles and the Stokes-Einstein relation.

 

(see S1 File for more details), such that they can cluster in the cytoskeletal network and also form large aggregates as found in experiments [28]. The exact position of the KAHRP interaction sites is based on the experimentally observed association properties of KAHRP, which have been found to be the following:

- the spectrin-actin-protein 4.1 junction [28, 44]

- $\alpha$-spectrin repeat 4 [45] and $\beta$-spectrin repeats 10–13 [29, 46, 47] which lie next to the tetramerization site (with $K_D = 50 \pm 15$ μM [29])

- the band 3 binding domain of ankyrin [47] (with $K_D = 1.3 - 1.8$ μM [48]).

Importantly, KAHRP-binding to ankyrin seems to occur together with binding to spectrin, but not at the ankyrin-spectrin binding site [47]. We therefore use the red beads in Fig 2A both to confine ankyrin to the membrane and to model KAHRP-binding to ankyrin/spectrin. The other binding sites on spectrin are shown in dark blue in Fig 2A and in the simulation snapshots presented later. Since the dissociation constant with these spectrin repeats is much larger than the one with ankyrin, we neglect the binding to the blue beads in the simulations. We also fix the KAHRP self-association interaction at a large enough value, such that small clusters can form in the cytosol. Therefore, we are left with two main parameters, the interaction energy between KAHRP and actin ($\epsilon_{KAHRP-actin}$) and the interaction energy between KAHRP and ankyrin ($\epsilon_{KAHRP-ankyrin}$). These binding energies are the depths of the respective Lennard-Jones potentials (Fig 2C). When interpreted as the binding free energy $\Delta G$, one can relate them to the dissociation constant through thermodynamic considerations [49]:

$$\Delta G = k_B T \ln \frac{K_D}{C}, \tag{3}$$

with C being a reference concentration and $K_d$ the dissociation constant.

## Actin dynamics

In contrast to KAHRP binding via a Lennard-Jones potential, we model the actin binding dynamics as explicit reactions, such that filaments can be formed as topologies in ReaDDy 2. When incorporating reactions into a Brownian Dynamics simulation, a distinction has to be made between macroscopic and microscopic reaction rates, where the diffusive process of finding a reaction partner is included in the macroscopic association rate $k^+$. The macroscopic and microscopic rates are related by [38, 50]

$$k^+ = 4\pi (D_1 + D_2) \left[ R_{12} - \sqrt{\frac{D_1 + D_2}{k^+_{micro}}} \tanh \left( R_{12} \sqrt{\frac{k^+_{micro}}{D_1 + D_2}} \right) \right], \tag{4}$$

where $D_{1/2}$ are the diffusion constants of the two reaction partners and $R_{12}$ is their separation.

Since the diffusion time scale is much faster than the reaction time scale, in order to obtain reasonable simulation times we use a factor $b$ that speeds up the elongation rate $\omega$, as introduced in [51], and indicate the scaled rates by a hat symbol:

$$\hat{\omega} = b\omega = b(k^+ C + k^-) = \hat{k}^+ \hat{C} + \hat{k}^-, \tag{5}$$

$$\hat{k}^+ = k^+ b_k, \quad \hat{C} = \frac{b}{b_k} C \text{ and } \hat{k}^- = bk^-, \tag{6}$$

where C is the concentration which determines the association and $k^-$ is the dissociation rate. In order to observe actin polymerisation within accessible simulation times, the scaling values

are chosen to be $b = 100000$, $b_k = 200$ and $\hat{C} = 500C$ unless otherwise stated. The parameters are chosen according to experimentally determined values if known. These values and the scaled value after accelerating the polymerization are given in S2 Table in S1 File. All relevant $K_D$ values are assembled in S3 Table in S1 File. The length regulation of the actin filaments is implemented through modelling capping proteins and introducing specific length dependent rules for the monomer dissociation. To prevent the filaments and hence the network from disassembling completely, the dissociation is stopped at two monomers introducing a boundary for the modelled process.

In order to mimic the attachment of tropomyosin which has the length of six monomers, the dissociation rates are chosen differently for filaments that are longer than six monomers or shorter. Specifically, the following rules are applied:

- G-actin dissociation at barbed and pointed ends:

   for $n > 6$ normal rates apply
   for $n \leq 6$ the rates are halved

- adducin dissociation rate always the same

- tropomodulin dissociation:

   for $n > 6$ the rate in absence of tropomyosin is used
   for $n \leq 6$ the rate with tropomyosin present is used

- association rates are not length dependent but depend on concentration

- tropomodulin and adducin block polymerization while attached.

We first set up a hexagonal array of spectrin tetramers (Fig 2D). In the z-direction the filaments initially form a sine curve with the red ankyrin attachment sites placed in the plane at $z = 10$ nm. This is necessary since the tetramer contour length of 205 nm is much longer than the typical actin-actin separation of 60 to 80 nm. The actin filaments at the junctional points are randomly oriented within the plane at $z = 10$ nm and the spectrin filament ends are placed at one of the six positions along both sides of the actin filament. Monomer positions are chosen randomly in the 3D space above the cytoskeletal layer. The network is then relaxed to its equilibrium state (Fig 2E). Simulation images were produced with VMD [52].

## Pair cross-correlation analysis

We use a two-dimensional pair cross-correlation analysis to quantify the co-localization of KAHRP with the other components of the network as recently used for super-resolution microscopy data [35, 53, 54]. First, we generate a distribution profile for each protein by adding a two-dimensional Gaussian distribution on their 2D projected coordinates obtained from the simulations. The standard deviation of the Gaussian distribution is set to 16 nm (corresponding to 38 nm full-width at half-maximum of the used STED-microscope). Using the distribution profile, we create an image for each protein with 1 nm pixel size. For two images, $I_R(x, y)$ (red channel for KAHRP) and $I_G(x, y)$ (green channel for another protein), the two-dimensional pair cross-correlation (PCC) between them is given by

$$C(r, r + \Delta r) = \frac{A_{image}}{\pi \Delta r(2r + \Delta r)} \sum_{\rho=r}^{\rho=r+\Delta r} \frac{\sum_{i,j} \sum_{m,n} I_R(x_i, y_j) I_G(x_m, y_n) \delta(|\mathbf{r_{ij}} - \mathbf{r_{mn}}| - \rho)}{\sum_{i,j} \sum_{m,n} I_R(x_i, y_j) I_G(x_m, y_n)} \quad (7)$$

where $A_{image}$ is the area of the image and $\Delta r = 6$ nm is the width of the radial bins.

## Results

### Simulation of spectrin and actin filaments

In order to test our computational setup, we first performed some calibration simulations related to the polymer character of the spectrin and actin filaments. We first examined the properties of a single spectrin filament and vary its length and subunit properties (Fig 3A). Excluded volume effects and an angular potential between three neighboring beads for bending stiffness are included as specified in the S1 File. We first simulated single spectrin filaments

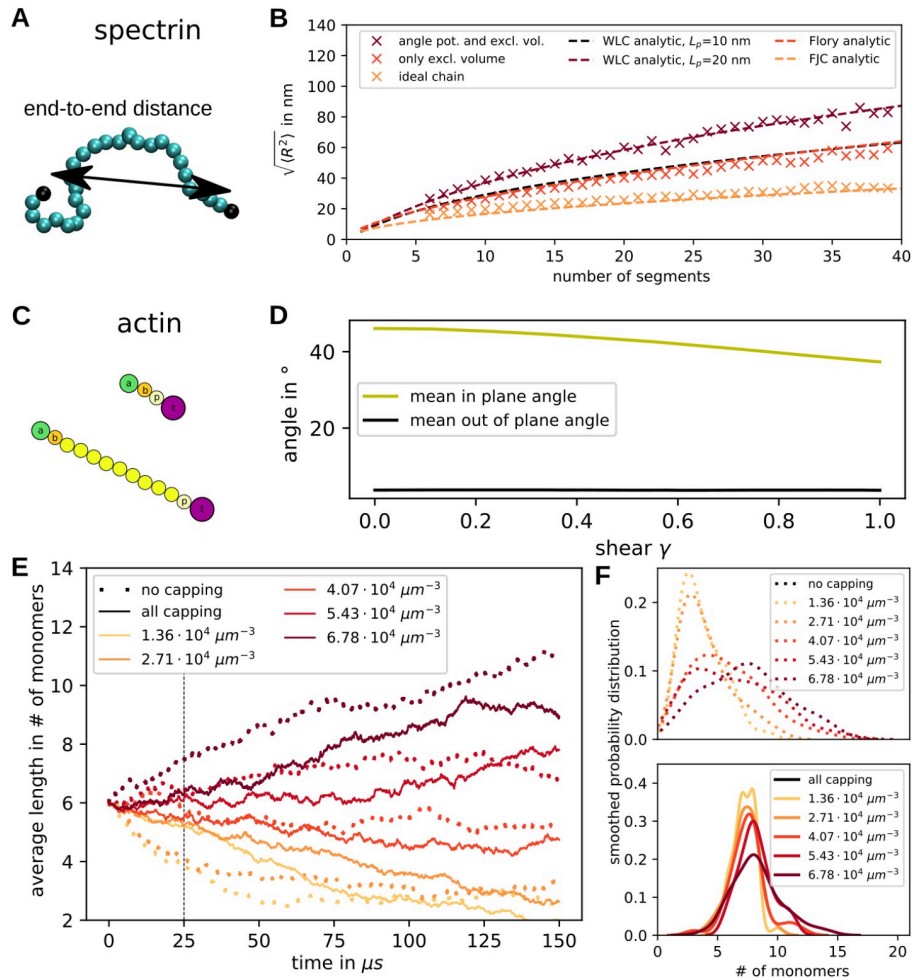

**Fig 3. Spectrin and actin filaments. (A)** Spectrin properties are tested by examining the end-to-end distance exemplified by the double arrow. **(B)** The root-mean-square value of the end-to-end distance is calculated for varying filament length. Simulations with an angle potential and excluded volume (wine) are compared to ones without angular confinement in red and simulations of ideal chains (i.e. no angle or excluded volume interactions) in orange. Additionally, we plot the analytic expression for the WLC (wine), the Flory theory (red) and the freely-joined chain (orange). **(C)** Capped actin filaments of different lengths. **(D)** The orientation of actin filaments within a network is quantified by the in-plane angle (orientation within the membrane) and the out-of-plane angle (angle in z-direction). These are plotted as a function of shear of the network. **(E)** The temporal evolution of actin filament average length is plotted for different initial values of G-actin concentration. The solid line shows the data from simulations containing capping proteins and the dotted lines show data from simulations without capping proteins. Each line corresponds to one simulated network with 46 actin filaments. The dashed vertical line indicates the time point for which the distributions are shown in F. **(F)** The smoothed probability distributions of the filament lengths at time 25 μs are shown for all concentrations. At the top the data is shown for simulations without capping proteins and at the bottom with capping proteins.

with an angle constant $k_{\text{angle}} = 4.28$ kJ mol$^{-1}$, calculated the time average of the end-to-end distance and compared the result to known polymer models in Fig 3B. Each point is an average over 20 million time steps of size 0.1 ns. The end-to-end distance is defined as the distance between the two end beads coloured black in Fig 3A. In Fig 3B three different types of filaments are compared, the full spectrin model in wine colour, a polymer without excluded volume but with an angle potential in red, and a polymer without any interactions except the harmonic potential connecting neighbours in orange. We see that the spatial extend of the polymer increases with the number of segments and is well-described by the polymer model appropriate for each case.

The full model corresponds to a WLC with a fitted persistence length of 20 nm, which is a factor of two larger than the theoretical prediction of the WLC-model because this model does not incorporate excluded volume effects. The red data points from simulations solely including excluded volume effects can be described by the Flory model for a polymer with self-interaction, but no persistence. The filaments without any interactions mimic the analytically calculated behaviour of a freely-jointed chain model. These results demonstrate that the polymer simulations in ReaDDy 2 give exactly the theoretically expected results for our test cases. This analysis also shows that a full spectrin filament containing 39 beads will have an end-to-end distance of 87 nm in equilibrium and it will be in a highly coiled state if not extended by force, thus explaining the typical distance between the actin junctional complexes and the finite thickness of the spectrin layer. Certainly the WLC-model does not reflect the full molecular complexitiy of the spectrin chains, as reflected e.g. by the Chinese finger trap model [8], but it represents well the typical dimensions and statistical properties of the spectrin chains in the network.

Going from the single spectrin filament to the full network (compare Fig 2E), we next examined the exact network thickness, which is defined as the distance in z-direction (vertical direction in the projection in S1(D) Fig) between the lowest and highest cytoskeletal bead. It can be seen that the spectrin tetramers are extended away from the bottom of the simulation box which can be thought of as the confining lipid bilayer. The simulated healthy network has an average thickness of 54.21 nm after equilibration. In S1(E) Fig the distribution of the different particles along the z-direction can be seen. The actin filaments are confined to a thin section near the bilayer, whereas the spectrin filaments are shown to be confined to a 58 nm thick plane which is less dense at its edges. The obtained values matches the experimentally determined thickness of approximately 50 nm of the dense cytoskeletal layer [15]. The experiments showed an additional 40 nm softer regime which is likely caused by defects in the network that are not present in the simulation.

In the next step we examined the modelled actin filaments, which appear as straight rods because of the much larger persistence length (Fig 3C). Therefore, we can examine their orientation within the cytoskeletal network as we apply a shear to the network with a shear rate of $3 \cdot 10^5$ s$^{-1}$. Initially, the filaments are set up parallel to the bilayer plane but randomly oriented within this plane. We can quantify this by two angles, the out of plane angle $\phi$ and the in plane angle $\theta$ (compare S1(A) Fig). Since we do not take into account the polarity for this analysis, the angles can vary between 0° and 90°. We now quantify how the angles change when going from the equilibrated reference state to the final state after shearing in x-direction of full-cytoskeleton simulations with $a = 88$ nm (see Fig 3D). It can be clearly seen how the average filament angle of all filaments in ten simulations is reduced as a function of shear. This means that the filament angles are not randomly distributed after the shearing, but they align in the shear direction. At the same time the angle in z-direction stays close to its small value. This angle is very close to zero because of the confining potential mimicking the attachment to the lipid bilayer. Histograms of this data can be seen in S1(B) and S1(C) Fig.

Depending on the G-actin concentration and the capping rates, the filaments can grow, shrink or stay at a constant length. The actin dynamics was sped up by a factor $b$ as explained in the Materials and methods section to reach reasonable modelling time scales. The change of actin filament lengths was examined by two types of simulations: one set was conducted without any capping proteins present ("no capping" in Fig 3E and 3F) and the other set contained capping proteins and special rules to mimic tropomyosin attachment in the simulations ("all capping" in Fig 3E and 3F). The average filament length and the filaments' smoothed probability distributions were examined for different concentrations of free actin monomers ($1.36 \cdot 10^4$ $\mu m^{-3}$ to $6.78 \cdot 10^4$ $\mu m^{-3}$) in Fig 3E and 3F, respectively. This range of values brackets the experimentally measured value of 0.36 $\mu$M for G-actin (compare S2 Table in S1 File) [17]. Although in principle possible at these concentrations, actin treadmilling was not observed in our simulations. We use a hexagonal set-up for the actin filaments as shown in Fig 2D and fix the lattice constant at $a$ = 88 nm.

In Fig 3E we see how the average filament length grows or shrinks depending on the initial concentration of G-actin. Capping does not completely suppress actin dynamics, but makes it much slower, such that the length stays close to the initial length of six monomers for a longer time. To see this in more detail, the filament length distribution was plotted for both cases at time point 25 μs in Fig 3F. The distributions for simulations without capping proteins are clearly much broader than the ones with capping proteins. In both cases the distributions for higher concentrations are spread out further because macroscopic association rates depend on concentration.

## Shear modulus

Next we set up a network of spectrin filaments which are connected via actin filaments at the junctional points and include various diffusing monomers as explained in the Materials and methods section. First, we examined the properties of the simulated cytoskeletal network without any additional malaria-associated alterations in order to validate our novel modelling approach. For examining the mechanical properties of the system, we set up a series of simulations (snapshots shown in Fig 4A–4C) mimicking a shear experiment with a shear rate of $3 \cdot 10^5$ $s^{-1}$, as it is common in the field [36], and extract the stress response as seen in Fig 4D–4H (an example of the shear simulation can be seen in S2 Movie). The curves show the mean of ten independent stress responses each.

In order to compare our full model including actin filaments, spectrin persistence and anchoring by band 3 to previous approaches [34, 36, 55], we set up a reduced system with actin filaments replaced by single particles (compare Fig 4A in contrast to the full model in Fig 4B), spectrin filaments with just excluded volume and free diffusion of the actin particles in the bilayer plane. The sheared networks have a size of three hexagonal units times four hexagonal units and vary in size depending on the lattice constants. We set up regular networks with actin particles positioned according to a uniform hexagonal network with lattice constant $a$ = 60–100 nm and compare their stress responses in Fig 4D. The network with $a$ = 100 nm displays a shear modulus of 10.92 μN m$^{-1}$ at low shear and 21.22 μN m$^{-1}$ at high shear as extracted from Fig 4D. We also see that for smaller lattice constant $a$ the strain-hardening behaviour of the network is reduced drastically in our model. Previous simulations which modeled actin filaments as one spherical bead also showed a monotonic increase in stress and yielded a modulus of $10 - 12$ μN m$^{-1}$ for small shear at similar shear rates [34, 36, 55]. Moreover this earlier work also predicted shear stiffening for shear values larger than 0.5. In Fig 4G we show how the stress changes when we introduce the spectrin persistence and the reduced diffusion of actins. Looking at the curve for $a$ = 100 nm, we see that for small shear the

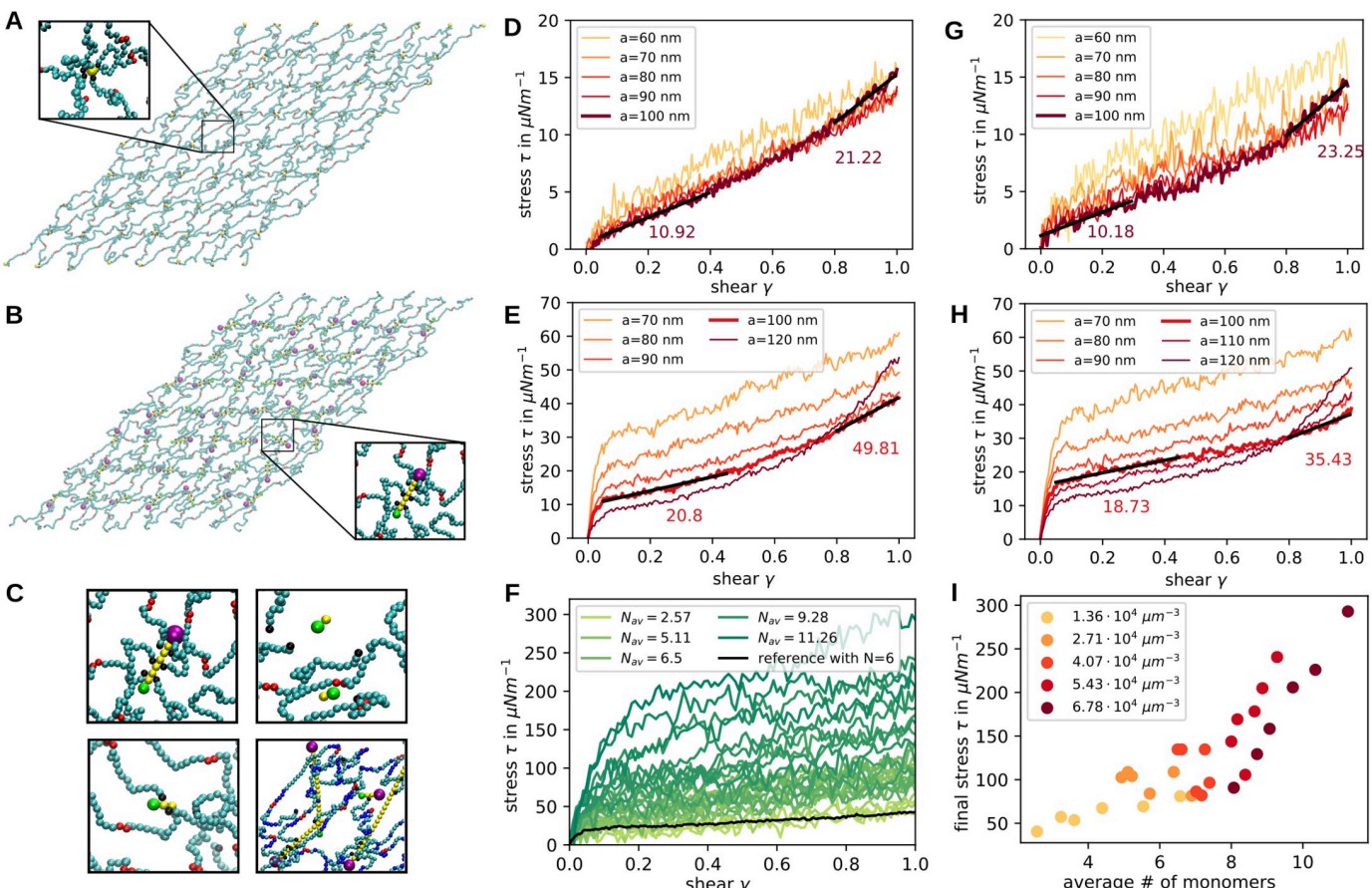

**Fig 4. Network shear response. (A)** Snapshot of the simulation at shear 1.0 in the x-direction. Junctional points are modelled as single actin particles. **(B)** Same as in **a** but now the junctions are modelled as full actin filaments. **(C)** Examples of actin filaments at different lengths within the network. **(D-H)** Stress is plotted against shear extracted from simulations with a shear rate of $3 \cdot 10^5 \, s^{-1}$. Each line corresponds to the average of 10 simulations. In **D and G** single particles are used as actin junctions whereas in **E and H** the proper actin filaments are implemented. In **D and E** the spectrin filaments are modelled without angle potentials and the anchoring sites are free to diffuse in the bilayer plane. In **G and H** the spectrin angle potential constant has a strength of 4.28 kJ mol$^{-1}$ and the anchoring sites possess a reduced diffusion due to anchoring in the bilayer. **(F)** The stress response to shearing is plotted for different average filament lengths as indicated in the legend. Sheared networks are taken from all different concentrations and time points of simulations with capping proteins. The black line is for a perfect network with exactly $N = 6$ actin beads per filament and serves as a reference. **(I)** The final stress from C is now plotted against the average filament length with the colours indicating the different initial concentrations as shown in the legend.

response is similar, but for larger shear rate the shear modulus increases to 23.25 μN m$^{-1}$. Considering the simulations with smaller lattice constant, the shear modulus is elevated at low shear.

In Fig 4E and 4H, the stress-shear response is shown for simulations containing explicit actin filaments rather than only actin beads. In Fig 4E the spectrin filaments do not have persistence and actin filaments can diffuse freely in the bilayer plane, similar to conditions in Fig 4D. In Fig 4H the response of the full model with persistence and anchoring is shown. The main new feature that appears in these stress responses is a stress jump at very low shear, similar to the elastic response of dense polymer networks [56]. After this initial response the shear modulus is given by 20.8 μN m$^{-1}$ and for high shear the value increases to 49.81 μN m$^{-1}$. Hence, the shear modulus is approximately doubled with respect to the previously examined networks which model the actin filaments as one particle. The stress jump is caused by the newly introduced actin filaments, where the reduced diffusion of these makes the jump even higher as seen in Fig 4H compared to Fig 4E. The shearing in these simulations was done at

shear rate $3 \cdot 10^5 \, s^{-1}$, matching the value of previous simulations. However, when we reduce this shear rate, the jump gets smaller and eventually vanishes (see S2 Fig). However, for smaller shear rates the stress magnitude is also reduced in general, as expected.

Because one of the main consequences of a malaria infection is actin mining, next we investigated networks with different actin filament lengths distributions. In Fig 4C the different possible outcomes are shown as simulation snapshots. The first panel shows a normal length actin filament, the second one shows completely detached short actin filaments, in the third panel the spectrin filaments are still attached to the very short actin and in the last panel abnormally long actin filaments are shown. Each sheared network is categorised in terms of the average actin filament length and the stress response is plotted in Fig 4F. As the colour of the curves varies form light to dark green, the average filament length grows. We see that the shear modulus and the initial jump increase with average filament length. To explore this in more detail, the final stress value at shear one is plotted in Fig 4I against the average filament length and the colours show the initial concentration of the simulation that the sheared network configuration was taken from. The higher stress value for longer actin filaments can be confirmed here, where there seem to be two regimes. For average filament lengths of less than seven monomers the fitted slope of stress against length is considerably smaller than for networks with average lengths longer than seven monomers. Overall we conclude that actin mining will decrease iRBC-stiffness.

As there is various evidence that the spectrin cytoskeleton is not perfectly connected [6], we also run several simulations with different connectivity of the network (Fig 5A). The underlying molecular processes might be very complex, thus we only considered four paradigmatic situations: complete removal of some spectrins, removal of some spectrins and stretching of others, removal of actin protofilaments and removal of complete hexagonal units around an actin protofilament. Simulation snapshots show how the resulting holes look like in the sheared networks (Fig 5B). In Fig 5C we see that the changes in spectrin affect the stress at high shear, where the stress is decreased for three spectrins per node and elevated for the long spectrins. In Fig 5D we see that removal of actin (either the filament only or the whole unit cell around it) reduces stress at all shear strengths and that overall shear modulus goes done, as expected from earlier theory [57] and simulations [41, 58]. We also studied the effect of a variable distribution of the distances between the actin junctional complexes by moving the actin particles away from their lattice sites by random displacements (S3 Fig). In this case we find a higher shear modulus and a stronger strain hardening behaviour in comparison to Fig 4D. This agrees with analytical calculations showing that random networks are stiffer [59] and can be explained by the contribution of the strongly elongated spectrin filaments.

## Cytoskeletal remodelling by KAHRP

Using our detailed model of the RBC cytoskeleton, we now are in a position to simulate the effect of KAHRP-binding during a malaria-infection. We introduce particles to the cytoskeletal system that possess the known binding properties of KAHRP, specifically the binding to other KAHRP particles, subunits of the spectrin filaments, ankyrin and actin protofilaments (explained in the Materials and methods). Because simulations with many interactions are very time consuming, a small membrane patch ($140 \times 242.48 \, nm^2$) with periodic boundary conditions and eight actin filaments is set up. The KAHRP particles initially exhibit a uniform random distribution and the lattice constant of the lattice is chosen to be $a = 88$ nm. Since KAHRP assembly is much faster than the actin dynamics, the effects of actin polymerizaiton on KAHRP assembly are negligible and hence we do not speed up the actin dynamics as in the last section.

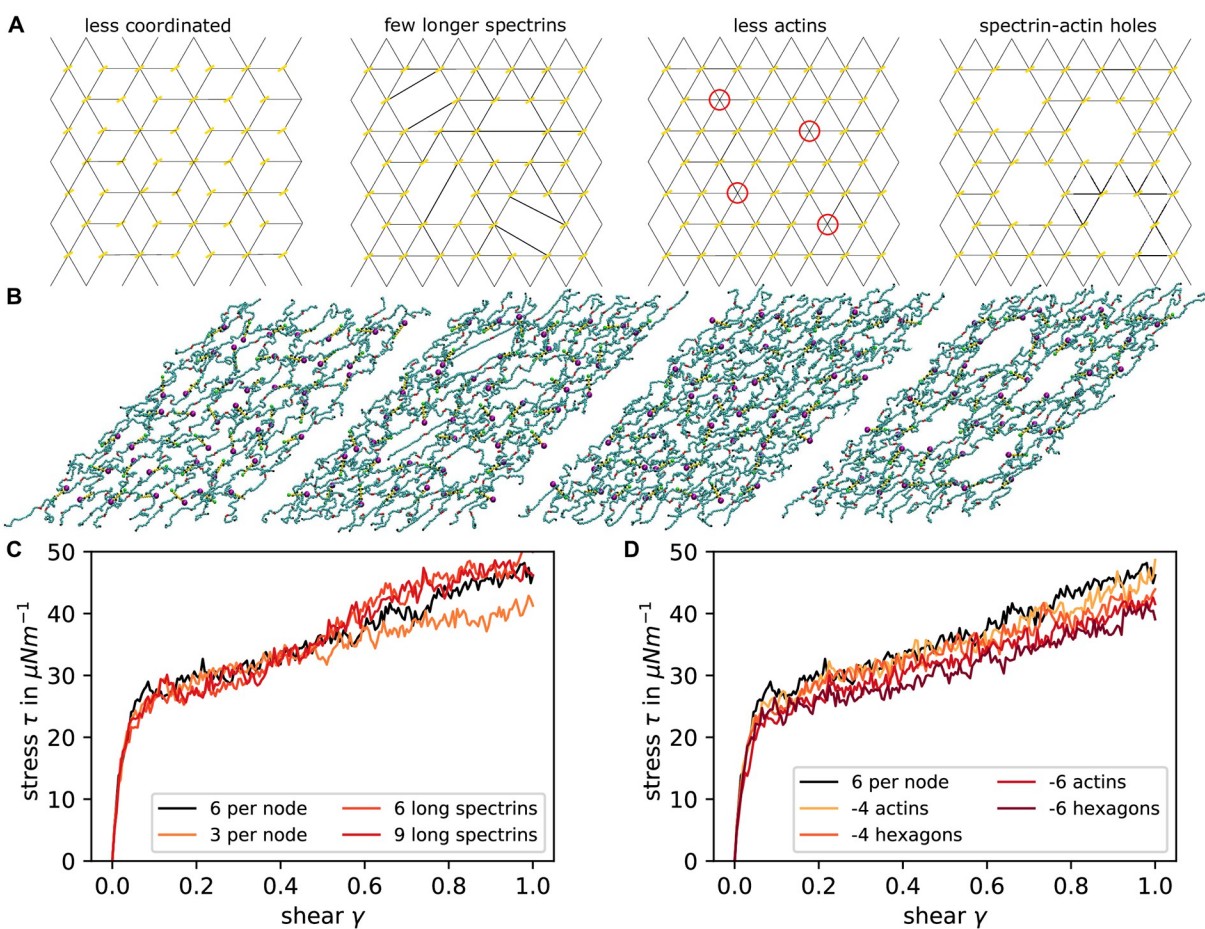

**Fig 5. Effects of network connectivity on shear response. (A)** Schematic representation for the network set-up. The yellow bars represent actin filaments and red circles indicate where an actin filament is missing. The black lines show the initialization positions of spectrin filaments. **(B)** Final configurations of the simulated networks at a shear of 1. The colors of the particles are the same as shown in Fig 2. **(C)** The stress-shear response is shown for networks with different connectivity. The black line corresponds to the perfect hexagonal network with six spectrins per node and the yellow line shows the response of a network with three spectrins per node. Additionally, some networks were simulated where four/six wholes where bridged with six/nine long spectrin filaments. **(D)** Here the effect of missing actins and spectrin filaments is analysed. Either only the actin nodes were taken away (-4/6 actins) or the whole hexagonal element including the spectrins (-4/6 hexagons). Again, this is compared to the perfect hexagonal lattice response in black.

In our simulations we varied the relative interaction strength of KAHRP with its different binding partners, the interaction strength itself, the actin filament length and the KAHRP concentration (simulation snapshots in Fig 6A and corresponding videos in S3–S6 Movies). In each case, we quantified the resulting KAHRP cluster. A cluster is defined as all particles whose centres are separated less than the diameter of a KAHRP particle plus a tolerance of 1.5 nm to the next KAHRP particle and examples are marked with circles in Fig 6A. We distinguish between free floating clusters, clusters attached to the actin junctions (red rectangles and one zoom-in) and clusters at the ankyrin junctions (black rectangles and one zoom-in). A cluster counts as attached if one of the cluster particles has a gap of less than 1.5 nm to one of the KAHRP-associating cytoskeletal components.

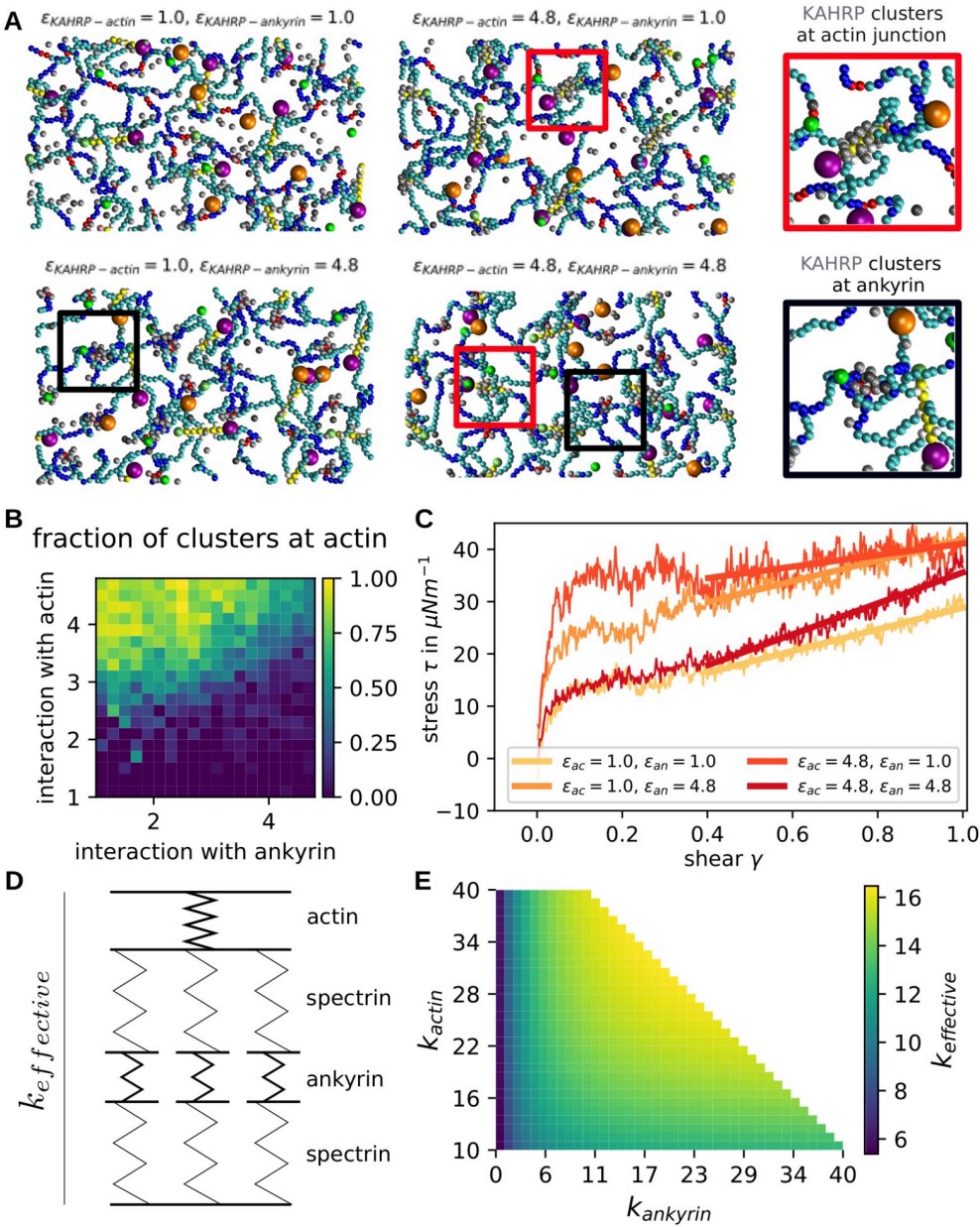

**Fig 6. Effect of KAHRP clusters in actin-spectrin network.** (**A**) Snapshots of equilibrated networks for the interaction energies indicated above each image. KAHRP particles are shown in grey and some KAHRP clusters are indicated by rectangles and shown as zoom-ins. G-actin particles are not shown here. (**B**) Fraction of clusters that are located at actin filaments in contrast to other cytoskeletal subunits. The value is calculated for different interaction energies with actin and ankyrin. (**C**) Shear response of KAHRP containing networks for different KAHRP positioning. (**D**) A simple spring model was set up to understand the shear-reaponse of the KAHRP-containing cytoskeletal networks. (**E**) The effective spring constant is plotted according to our spring model.

Our first finding is that when the interaction strength $\epsilon$ is too small compared to the thermal energy, most KAHRP particles will stay separated and detached from the cytoskeleton. In simulations with all interaction energies equal, we found that there is a sharp transition between mostly no clusters present below $\epsilon = 2\,\mathrm{k_BT}$ and similarly sized large clusters for $\epsilon \geq 2$

$k_B T$ (compare S4 Fig). These results essentially demonstrate that the energy of clustering competes with the entropy of diffusion in our dynamical simulation.

Next we varied the interaction energy between KAHRP and actin ($\epsilon_{KAHRP-actin}$) and the energy between KAHRP and ankyrin ($\epsilon_{KAHRP-ankyrin}$), each between 1.0 and 4.8 $k_B T$. At the same time we fix the interaction energy with KAHRP itself and the spectrin beads at $\epsilon_{KAHRP-KAHRP} = \epsilon_{KAHRP-spectrin} = 1.0\ k_B T$. For each parameter set we can then extract the properties of KAHRP clusters. Specifically we can calculate the fraction of all KAHRP clusters that are positioned at actin filaments (Fig 6B). The data corresponds to the final configuration of a 200 μs simulation. We confirmed that this is the equilibrium configuration by examining the potential energy of the system (not shown here). It can also be seen by looking at the time evolution of the fractions and sizes (shown in the S4 Fig). Snapshots for selected parameters, corresponding to the four diagram corners in Fig 6B are shown in Fig 6A. We see that the interaction with actin needs to be larger than the interaction with ankyrin to reach a high fraction of actin associated clusters (top left corner in Fig 6B). If the interaction is equally strong but large enough, we see a mixed distribution of clusters. Note that no clusters develop if both parameters very small. The size of the clusters can be varied by changing KAHRP concentration (S5 Fig).

To examine the effect of the emerging KAHRP clusters on the mechanical properties of the network, we took the final configurations from different parameter choices discussed in the last paragraph and applied a shear to the network. To do so, the periodic boundary conditions were removed and the network repeated nine times to obtain a larger patch and hence better statistics. Shearing was done at a rate of $3 \cdot 10^5\ \text{s}^{-1}$. Specifically, we sheared networks with the four extreme cases of KAHRP cluster formation corresponding to the snapshots in Fig 6A: no clusters ($\epsilon_{KAHRP-actin} = 1.0 k_B T$ and $\epsilon_{KAHRP-ankyrin} = 1.0 k_B T$), mostly ankyrin associated clusters ($\epsilon_{KAHRP-actin} = 1.0 k_B T$ and $\epsilon_{KAHRP-ankyrin} = 4.8 k_B T$), mostly actin associated clusters ($\epsilon_{KAHRP-actin} = 4.8 k_B T$ and $\epsilon_{KAHRP-ankyrin} = 1.0 k_B T$) and clusters at both junctions ($\epsilon_{KAHRP-actin} = 4.8 k_B T$ and $\epsilon_{KAHRP-ankyrin} = 4.8 k_B T$). The different stress responses are plotted in Fig 6C and one clearly sees that for clusters at both junctions only the slope is increased and hence the shear modulus, whereas the initial step is higher for the formation of clusters only at one junction type, with the step being the largest for actin-associated clusters. Besides the step the shear modulus is higher for the ankyrin associated clusters.

Previously we saw that longer actin filaments lead to an increased initial step. Therefore, the clustering of KAHRP at one junction type, which leads to relatively large junction points in the network, results in a higher initial step. This effect is larger for actin associated clusters, since the actin forms a larger junction from the start. It seems like networks with clusters at both junction types do not show this effect. This could be explained by the fact that the simulations are run at constant KAHRP concentrations and thus the resulting clusters are smaller because they are distributed over different binding sites. Although the initial jump is smaller, the slope at larger shear is larger. In order to understand this aspect better, we used a simple spring model as shown in Fig 6D which uses different springs for actin/ankyrin junctions and spectrin filaments. The spectrin springs are arranged in parallel to mimic the attachment of several spectrins to one actin junction. The resulting effective spring constant of the simple model was calculated for different actin and ankyrin spring constants and plotted in Fig 6E with the spectrin spring constant fixed at 5 in non-dimensionalised units. From this plot we see that we expect the largest spring constant and hence shear modulus for a stiffening of both junctions at the same time, as seen in the simulations of Fig 6C. When keeping $k_{ankyrin}$ fixed at a low values, which corresponds to clusters at only the actin junctions, we expect nearly no increase in shear modulus as seen in Fig 6E on the left side for a change along the y-axis. In our shear experiments the slope stayed very low as expected here. Finally, for the case of ankyrin associated clusters, we see that the effective spring constant does change when going along the x-axis

of Fig 6E. Note that we start the plot at an actin spring constant of 10 in order to take into account that the actin junction is the stiffest element of the network without KAHRP present. This explains the observed higher shear modulus for ankyrin associated clusters in contrast to actin associated clusters.

## Pair cross-correlation analysis

Finally we used pair cross-correlation (PCC) analysis to connect our simulation results with the experimental results from our very recent two-color super-resolution microscopy study of dynamic KAHRP localization within the iRBC cytoskeleton [35]. Fig 7A shows a sample image of fluorescently labeled KAHRP and ankyrin in an exposed RBC membrane skeleton and Fig 7B and 7C show the resulting PCCs, which can be interpreted as the probability of finding two signals at a given radial distance. For co-localizing signals, the highest PCC is observed at a zero distance and for large separations between signals, the highest PCC shifts to a finite distance. Fig 7B shows that the PCC profile between KAHRP and ankyrin shifts from a maximum value at zero up to 16 hours to a flat profile at 20 hours and eventually to a maximum value at a finite distance ($\approx$ 110 nm) at 28–36 hours post malaria infection. In contrast, the PCC between KAHRP and the N-terminus of $\beta$-spectrin shows the maximum value at zero throughout, as shown in Fig 7C. This suggests that during the malaria infection, KAHRP relocates and moves away from the ankyrin complexes and towards the actin complexes.

To compute PCCs from our simulations, we generated a spatial distribution profile for each protein as described in the Materials and methods. Fig 7D shows the generated images for KAHRP in red and ankyrin in green colour. We first note that these images should not be compared directly to the experimental images in Fig 7A, because the scale is much smaller and the regularity is much higher due to the assumed hexagonal lattice in the computer simulations. Nevertheless we can calculate PCCs from them which can be compared to the experimental data. In Fig 7E and 7F we show the calculated PCCs for KAHRP with ankyrin and for KAHRP with the N-terminus of spectrin located at actin junction, respectively, for different values of the binding energy between KAHRP and actin as indicated by the color bars. The PCC of KAHRP with ankyrin decreases with the increase in its binding energy with actin, and the maximum PCC shifts from zero distance to a finite distance. This distance corresponds to the separation between actin and ankyrin junctions ($\approx$ 55 nm) in our simulated RBC-networks. However, the experimental PCC (shown in Fig 7B) shows a larger separation distance between the two junctions. This suggests that the RBC networks in the experiments are highly stretched. In contrast to the PCC with ankyrin, the highest PCC of KAHRP with the N-terminus of spectrin remains at a zero distance. Further, for strong KAHRP-actin binding, we observe a second peak at larger distances due to the KAHRP's correlation with N-terminus of spectrin present at the next nearest neighbour site of the RBC network.

Next, we varied binding energy between KAHRP and actin and KAHRP and ankyrin to systematically identify the regions in the parameter space where such behavior exists. In Fig 7G, we plot the PCC at zero distance for different binding energy values and mark the cases with finite distance peak with white dots. For short actin filaments, KAHRP-ankyrin PCC shows a maximum at zero distance in large regions of the parameter space, except in areas with very high KAHRP-actin binding. In contrast, the KAHRP and N-terminus of spectrin PCC shows the maximum at a finite distance for high KAHRP-ankyrin and low KAHRP-actin binding energy. This is because KAHRP preferentially binds to the ankyrin junction, thus not at the actin junction where the N-terminus of spectrin is located. At an increased actin length (Fig 7H for 36 nm and Fig 7I for 48 nm), we observe increased regions in the parameter space where the KAHRP-ankyrin PCC shows a maximum at a finite distance for high KAHRP-actin

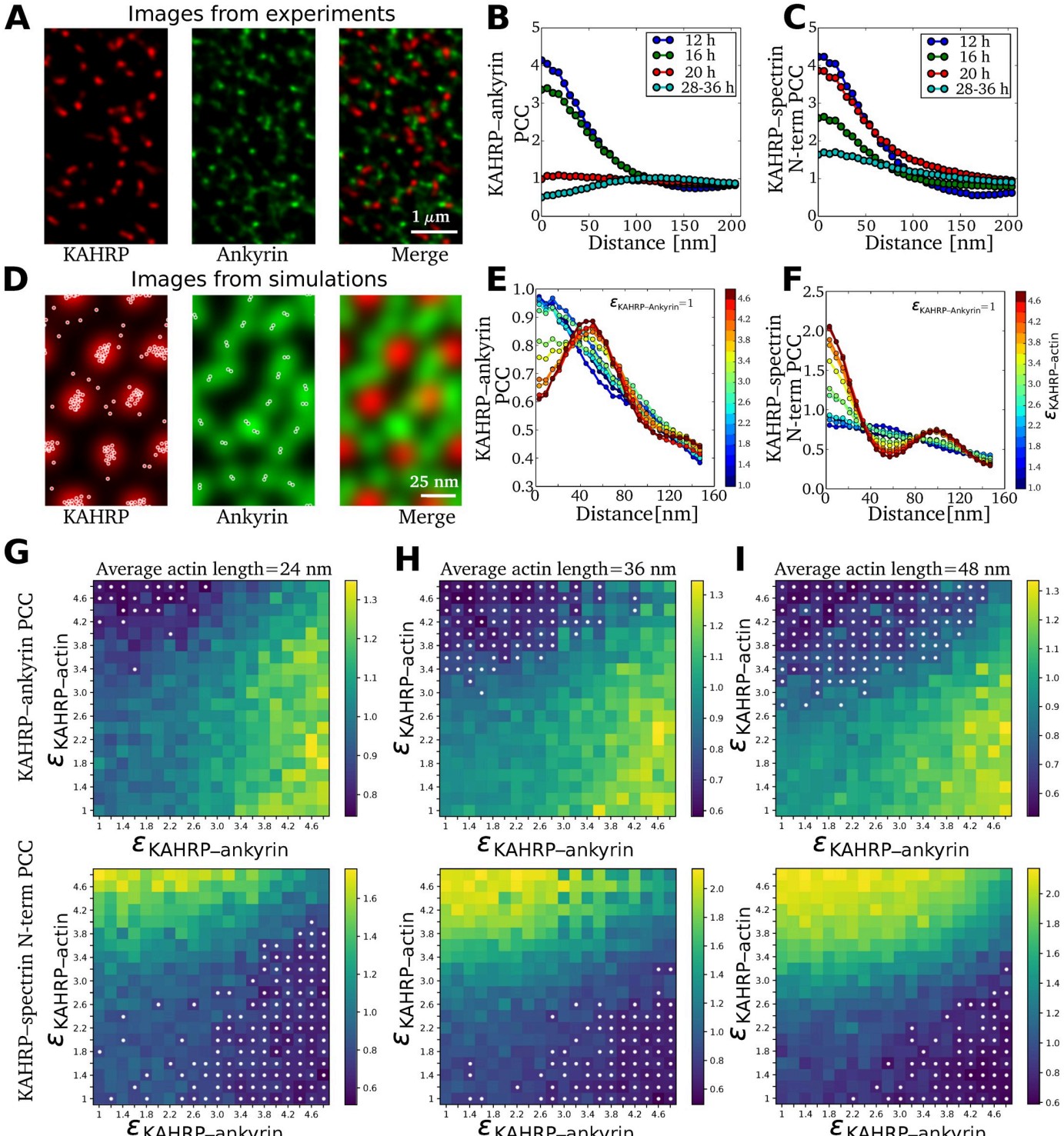

**Fig 7. Comparison of experimental and simulated pair cross-correlations. (A)** STED images of an exposed RBC membrane at the trophozoite stage (28–36 hours post malaria infection). Red and green fluorescence signals correspond to KAHRP and ankyrin sites on the RBC membrane. **(B)** Pair cross-correlation (PCC) between KAHRP and ankyrin computed using the two-color images obtained at different hours post malaria infection. **(C)** PCC between KAHRP and N-terminus of $\beta$-spectrin. Images and PCC values in (A,B) are taken from [35]. **(D)** KAHRP (red) and ankyrin (green) fluorescence signal is constructed from location points obtained from simulations (overlaid white points). **(E)** PCC between KAHRP and ankyrin is computed from two-color images generated from simulations for different binding energy between KAHRP and actin as indicated by the color bar. **(F)** Same as (E) for the KAHRP and N-terminus of spectrin PCC. **(G)** The map shows PCC values at zero distance for KAHRP-actin pairs (top panel) and KAHRP-ankyrin pairs (bottom panel) at different binding rates between

KAHRP and ankyrin/actin sites for average actin filament length 24 nm. The white dots mark the cases where a peak is observed at a finite distance. **(H)** and **(I)** Same as (G) for average actin filament length 36 nm and 48 nm respectively.

and low KAHRP-ankyrin binding energy. These results suggest that the temporal change in KAHRP-ankyrin PCC during the malaria infection can be explained both by differences in the binding affinity between KAHRP and actin or by changes in actin filament length. We also investigate the effect of KAHRP concentration on the pair cross-correlation values. The increase in KAHRP concentration decreases the region where the PCC maximum at a finite distance is observed (S6 Fig).

## Discussion

In order to theoretically analyse the dramatic transformation of the RBC cytoskeleton during a malaria infection, here we have developed a particle-based simulation framework that incorporates both the structure and dynamics of actin filaments and the different known binding sites of KAHRP, which is the most important parasite-derived factor for this transformation. Our coarse-grained Brownian dynamics simulations allow for sufficiently long and large simulations to measure the effective cellular shear modulus as a function of the key molecular processes. The shear modulus in turn is the central quantity to predict how iRBCs will move in the blood flow and through the interendothelial slits of the spleen. Our work is scale-bridging in the sense that it connects the spatial coordination of essential molecular processes, that are increasingly investigated with super-resolution microscopy [18, 31, 35], with cellular properties like the shear modulus, which has been measured before with different experimental techniques [23, 24]. Although a complete simulation of the cytoskeletal dynamics during a malaria infection is currently not possible due to missing experimental information, our simulation study helps to better understand some of the molecular mechanisms that underlie this process.

Our first main result is the observation that including explicit actin filaments into the simulations leads to two regimes during shearing, namely a fast and stiff response corresponding to their reorientation and a slow and less stiff response corresponding to the reorganization of the spectrin network. This latter response shows strain stiffening, as expected, and agrees quantitatively with results from similar coarse-grained computer simulations in which the actin protofilaments have been modelled as particles [34, 36, 37]. These results also agree well with values for the shear modulus extracted from experiments with optical tweezers [23] or diffraction phase microscopy [24]. Regarding the identification of a fast and stiff initial response, we note that it seems challenging to demonstrate this effect in experiments, because it would require very high time resolution and good preparation of the initial state. Yet we believe that this is an interesting new effect that might play a role in the physiological context. In the future, a more detailed investigation should also include the possibilities of actin protofilaments flipping out of the plane of the membrane [60, 61] and of unfolding of spectrin repeats [62, 63]. Both of these processes could yield extra length that would smear out the effect of the rapid ordering of the actin protofilaments. We finally note that whole cell simulations with molecular details are required to decide how this effect is averaged over the complete area of the RBC [37].

Our simulations not only modeled for the first time the actin component as explicit filaments, they also included their dynamical nature, allowing for them to shrink or grow. We found that by introducing length-dependent capping rates with dissociation constants matching experimental observations, filaments could be held stable over a longer time period than without capping proteins present. In order to improve the model in this respect, the rules for tropomyosin attachment should be refined. By testing different G-actin concentrations, filament networks of various average filament lengths were produced and their response to shear

could be determined. We found that shorter filaments lead to a reduced stress because the network looses connection points, suggesting that the actin mining by the parasite should reduce the shear modulus if no other processes were present.

The most important remodelling process triggered by the parasite is the clustering of the protein KAHRP, which has known binding sites to different components of the membrane skeleton. By introducing new particles into the computer simulations that possess the properties of KAHRP molecules, the KAHRP cluster formation could be analysed in detail, yielding a different cluster positioning within the cytoskeleton depending on the relation between the interaction strengths of KAHRP with its binding partners. In general, we found that KAHRP clustering, which eventually corresponds to knob formation, leads to an increase of the shear modulus, in agreement with earlier computer simulations that focused on the mature knobs [34]. By varying the different interaction strengths (which could also be interpreted as different values of $K_D$), we could predict the effect on the shear modulus for different scenarios of KAHRP clustering. In principle, the recently observed KAHRP relocalization from ankyrin to actin complexes could result from different effects, including changes in binding affinity due to phosphorylation, increase in concentration and structural changes to the target sites (in particular the remodelled actin junctional complexes). Our simulations showed that the first mechanism is the most likely candidate, because its effect resembles the experimentally observed dynamics. Our conclusion is in agreement with experimental observations that during the intra-erythrocytic cycle the phosphorylation and/or acetylation pattern of KAHRP are changing [64, 65]. While the effect of KAHRP concentration seemed to be small in the simulations, dynamic changes of actin filament length did play some role, as they could increase the number of binding sites at the knobs.

In order to compare simulations and experiments, we used a pair cross-correlation (PCC) analysis. We found that indeed the experimentally observed time course can be recapitulated by increases in the binding affinity between KAHRP and actin. Because our computer simulations work on a better resolved scale and assume a regular lattice, the corresponding PCCs show more structure than the experimentally measured ones, similar to earlier work on healthy RBCs [18], but the qualitative agreement is good enough to conclude that dynamical changes in KAHRP binding affinities are the most likely mechanism for the observed spatial changes.

Increases in the shear modulus are essential for RBC quality control in the spleen, because stiff RBCs cannnot squeeze anymore through the interendothelial slits and then are removed by macrophages. This mechanism does not only apply to old RBCs, but also to iRBCs, and is thought to be the main reason why iRBCs have evolved cytoadhesion to prevent clearance by the spleen [2]. Interestingly, this process also seems to be important for RBC-maturation, because reticulocytes become softer before going into the circulation [41]. Here we have shown that actin mining by the parasite, which it uses to build new transport pathways to the membrane (Fig 1B) [20], in fact decreases shear modulus by leading to shorter protofilaments (Fig 4I) and holes in the network (Fig 5D). At the same time, however, this makes space for longer spectrin connections and the formation of KAHRP-based clusters and eventually the knobs, which both increase shear modulus (Figs 5C and 6C, respectively). This agrees with earlier results that knob formation is the main driver for stiffening [34], but gives more quantitative information on the exact role of actin and KAHRP during this process. Overall the picture emerges of a multifaceted process that requires regulation. In particular, our results suggest that parasite-controlled affinity changes are required to explain the observed relocalization of KAHRP. We speculate that these are effected by posttranslational modification, in particular by tyrosine phosphorylations of KAHRP. Together with the molecularly still unclear perturbation of actin capping, this might be the most important molecular change required for the

observed time course of the RBC cytoskeletal dynamics during a malaria infection and should be a focus of future research in this field.

Because we simulate a small patch of the RBC skeleton with large molecular detail, we do not address the role of membrane fluctuations. In the future, this might be addressed e.g. within the multiscale framework OpenRBC [37, 66]. To address this issue in ReaDDy 2 [38], a particle-based membrane model had to be implemented there. Recently it was shown with continuum approaches that a similar conclusion is to be expected, namely that knob formation will dominate over network degradation [67]. Membrane fluctuations in RBCs do result not only from thermal activation, but also from active processes consuming ATP [57, 68–70]. Active fluctuations of the membrane might arise because ATP-binding to actin junctional complexes leads to spectrin dissociation, possibly mediated by the transmembrane protein GPC or protein-4.1 that enhances the actin-spectrin binding. In the case of iRBCs, it has been shown that ATP is released via parasite-induced new permeation pathways in the red blood cell membrane [71], but that the parasite maintains the ATP-equilibrium by exporting ATP into the erythrocyte cytosol [72]. This suggests that the changes in cell mechanics of iRBCs mainly result from parasite-induced cytoskeletal remodelling as discussed here. However, future experimental investigations should check this expectation by studying the effect of ATP-depletion on the cytoskeletal time course during a malaria infections.

In summary, our newly developed computer simulations provide an exploratory tool to investigate the different points of attacks that the malaria parasite might use to remodel the membrane skeleton of iRBCs to achieve favorable flow and adhesion behaviours in the vasculature. Here we focused on spatial processes on the molecular scale that directly translate into the mechanical properties relevant for cell movement in hydrodynamic flow. Our finding that changes in actin stabilization and KAHRP affinity might be the central features of interest suggests that also other more biochemical processes might play a role, including oxidative stress. Before these important processes can be included here, more experimental evidence is required to guide such a modelling approach. We finally note that future work should also become more three-dimensional. Although our simulations are three-dimensional in regard to the spectrin network and reproduce well the known thickness of this layer, they did not consider three-dimensional changes for the actin protofilaments [60], the formation of the long actin filaments connected to the Maurer's clefts [20] or the spiral scaffold underlying the knobs [30, 31, 35]. As a long-time perspective, it would be very desirable to develop also a spatial model for these more three-dimensional processes.

## Supporting information

**S1 File. Details on the modelling procedure and tables. S1 Table. Parameters for the repulsion potentials**. The value for spectrin is chosen to match previous simulations in Ref. [36]. **S2 Table. Summary of actin dynamics rates**. Rates and concentrations relevant for the actin dynamics are collected here. The real value is given with its reference where applicable and the value scaled in order to produce observable actin dynamics within an accessible time period. The scaling factor b was set to $10^5$. **S3 Table. Experimental values for $K_D$.** Dissociation constants are listed, that have been found by various groups for different protein (fragment) pairs. The three parts of the table show interactions of host cytoskeletal proteins, interactions with KAHRP and interactions with the cytoplasmic domain of PfEMP1 from top to bottom. (ZIP)

**S1 Fig. Network configuration. (A)** Definition of actin filament angles in space. Both angles can vary between 0 and 90˚ as the filament polarity does not matter. **(B)** Distribution of the out of plane angle $\phi$ at the initial and final time point of the simulation. The data from 10

independent runs of a network with 46 actin filaments each is used. **(C)** Same as in b but for the orientation within the plane of the bilayer quantified with angle $\theta$. **(D)** Side view of a simulation snapshot with a confining potential for the actin filaments near the bottom of the simulation box. Actin filaments are shown in yellow and spectrin filaments in cyan and blue. Diffusing monomers are important for the actin polymerisation. **(E)** Distribution of different particle types given as distance from the bottom simulation boundary mimicking the lipid bilayer. Actin filaments and spectrin filaments are considered separately.
(TIF)

**S2 Fig. Shear rate effects.** Effect of different shear rates on the stress in the modelled network is shown for different lattice constants. **(A)** The shear rate was set to $1.5 \cdot 10^5 \, s^{-1}$. **(B)** The shear rate was set to $5.0 \cdot 10^4 \, s^{-1}$. **(C)** The shear rate was set to $1.7 \cdot 10^4 \, s^{-1}$.
(TIF)

**S3 Fig. Effects of random displacement on shear response.** For these shear simulations the position of the actin particles was randomly displaced from their hexagonal lattice site. The extracted stress is shown here for different lattice constants.
(TIF)

**S4 Fig. Formation of actin-associated clusters. (A-C)** The fraction of clusters near actin filaments is plotted over time and distinct conditions as explained hereafter. **(D-F)** Average size of cytoskeleton attached clusters is plotted for the same conditions. In **(A)** and **(D)** the relative strength between the KAHRP-actin and the KAHRP-spectrin interaction is varied. In **(B)** and **(E)** the interaction strength is varied and in **(D)** and **(F)** the KAHRP concentration. For each of the three columns the other two parameters are kept fixed at the value indicated by the red horizontal line. Each data point corresponds to the average of three simulations.
(TIF)

**S5 Fig. Effects of KAHRP concentration on cluster size.** The map shows the average size of different KAHRP clusters, i.e., free, actin-associated, and ankyrin-associated for different binding energy between KAHRP and ankyrin/actin junctions and for different concentrations of KAHRP.
(TIF)

**S6 Fig. Effect of KAHRP concentration on pair cross-correlation.** The map shows PCC at zero distance for KAHRP-actin pairs (top panel) and KAHRP-ankyrin pairs (bottom panel) for different KAHRP concentrations shown in S4 Fig.
(TIF)

**S1 Movie. Simulation of the healthy RBC cytoskeleton.**
(AVI)

**S2 Movie. Movie of RBC cytoskeleton shear simulation.**
(MP4)

**S3 Movie. Simulation of the RBC cytoskeleton for low KAHRP-actin and KAHRP-ankyrin binding energy.**
(AVI)

**S4 Movie. Simulation of the RBC cytoskeleton for high KAHRP-actin and low KAHRP-ankyrin binding energy.**
(AVI)

**S5 Movie. Simulation of the RBC cytoskeleton for low KAHRP-actin and high KAHRP-ankyrin binding energy.**
(AVI)

**S6 Movie. Simulation of the RBC cytoskeleton for high KAHRP-actin and KAHRP-ankyrin binding energy.**
(AVI)

## Author Contributions

**Conceptualization:** Julia Jäger, Pintu Patra, Michael Lanzer, Ulrich S. Schwarz.

**Data curation:** Pintu Patra, Cecilia P. Sanchez.

**Formal analysis:** Julia Jäger, Pintu Patra.

**Funding acquisition:** Michael Lanzer, Ulrich S. Schwarz.

**Investigation:** Julia Jäger, Pintu Patra, Cecilia P. Sanchez.

**Methodology:** Julia Jäger, Pintu Patra, Cecilia P. Sanchez.

**Project administration:** Michael Lanzer, Ulrich S. Schwarz.

**Resources:** Michael Lanzer, Ulrich S. Schwarz.

**Software:** Julia Jäger, Pintu Patra.

**Supervision:** Michael Lanzer, Ulrich S. Schwarz.

**Validation:** Julia Jäger, Pintu Patra, Cecilia P. Sanchez.

**Visualization:** Julia Jäger, Pintu Patra.

**Writing – original draft:** Julia Jäger, Ulrich S. Schwarz.

**Writing – review & editing:** Julia Jäger, Pintu Patra, Michael Lanzer, Ulrich S. Schwarz.

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
