## [Decision Letter · Decision Letter 0]

16 Nov 2021

Dear Prof. Dr. Schwarz,

Thank you very much for submitting your manuscript "A particle-based computational model to analyse remodelling of the red blood cell cytoskeleton during malaria infections" for consideration at PLOS Computational Biology.

As with all papers reviewed by the journal, your manuscript was reviewed by members of the editorial board and by several independent reviewers. In light of the reviews (below this email), we would like to invite the resubmission of a significantly-revised version that takes into account the reviewers' comments.

We cannot make any decision about publication until we have seen the revised manuscript and your response to the reviewers' comments. Your revised manuscript is also likely to be sent to reviewers for further evaluation.

Sincerely,

Alison L. Marsden

Associate Editor

PLOS Computational Biology

Jason Haugh

Deputy Editor

PLOS Computational Biology

Reviewer's Responses to Questions

**Comments to the Authors:**

Reviewer #1: The authors present a detailed and interesting study of the effects of the malaria proteins on the RBC cytoskeleton shear stiffness and organization. I think this is important, and the tools are appropriate. I have some comments that need addressing before publication

1) A few English editing is needed:

Line 13: "the membrane and there incorporation into the existing surface"

Line 39: "finding that malaria parasite mine actin from the junctional complexes, which then is used to"

2) All the proteins denoted in Fig.1 should be explained in the caption.

3) Line 145: "Since the dissociation constant with spectrin repeats is much larger than the one with ankyrin, we neglect the binding to the blue beads and consider binding to ankyrin as binding to the red beads

in Figure 2A. Note that we do not model ankyrin explicitly but KAHRP binds directly to the ankyrin binding site on the spectrin molecule."

But this way the KAHRP will bind to spectrin on the simulated membrane, and in reality it can displace the spectrin from its membrane anchoring proteins and therefore remain only attached to the ankyrin/actin on the membrane, and in the process dissociate the spectrin filaments. Why dont they consider this possibility ?

4) Why do they only consider themodynamic equilibrium interactions and no ATP-induced dissociations of the spectrin-actin junction ? or spectrin tetramer dissociations ?

5) Can the shear response calculated in Fig.4 be compared in more detail to experimental results ?

6) The authors consider fully-conneced spectrin networks, but recent work suggests that the spectrin network is not fully connected:Li, He, et al. "How the spleen reshapes and retains young and old red blood cells: A computational investigation." PLOS Computational Biology 17.11 (2021): e1009516.‏

A large percentage of dissociated filaments in the normal RBC network was also discussed theoretically in:

Gov, N. S., and S. A. Safran. "Red blood cell membrane fluctuations and shape controlled by ATP-induced cytoskeletal defects." Biophysical journal 88.3 (2005): 1859-1874.

7) Fig.5a: please define the KAHRP molecules, what is their color ? provide zoom-in images of the clusters that they make in each of the cases.

8) I dont see that in any of the main results there is a crucial role for the "actin mining" that was promoted in the abstract. Maybe this is not such an important process ?

9) In he context of Fig.6: They also do not consider that as time progresses there can be an accumulation of changes to the spectrin network that will also affect the localization of the KAHRP molecules, such as more spectrin-actin dissociations which will make more actin available for KAHRP binding.

Reviewer #2: The authors developed a sophisticated red cell membrane model to study its dynamics in malaria-infected cases with KAHRP proteins. It is an interesting study, which might be helpful for understanding better the red cell mechanics and malaria. Since KAHRPs interact with actin,s the polymerization process might be important, which is captured in this study for the first time.

The major concern is that some basic elements might not be right in this model, in spite of many advanced features, such as actin polymerization.

Here are the major questions:

1. The spectrin tetramer have two strands, but it is modeled as one. Even ankyrins will be doubled if it is modeled as two than one. Is it possible to model it as two strands to see the difference? There seems no particular technical challenges to do so, especially considering that more complicated features such as polymerization are modeled.

2. Many recent studies showed that spectrins might not behavior as random coils at all, such as the Chinese Finger Trap Model by McKnight group, since many EM images showed the spectrin maintains straight-line shape rather than random-coil shape in the native state. Any discussion on this? If the spectrin model is wrong, no matter how many advanced features about actin are added, the model might not be realistic.

3. One major assumption is that actins constantly polymerize in red cells, as Fowler’s group showed so. But it is still a controversial topic. At least, the authors should discuss why they think actin polymerize in red cells with these capping proteins, because for a long time, actin filaments in red cells are assumed to have constant lengths. A discussion of Fowler group’s results and alternative opinions might behelpful.

4. Similarly, how these polymerization constants are determined in red blood cells? Table S2 show concentrations of G actin, its two capping proteins (adducin and tropomodulin) in red cell from Gokhin et al.. The authors can search the proteomic data of red cells published recently by Gautier et al. to see whether their assumed concentrations are reasonable. In Table S2, it seems the scaled values of rates are obtained to match K_D. Which K_D? What are references for these K_D?

5. Since the spectrins are modeled as entropic springs, sufficient sampling is the key to capture its free energy. Although Fig. 3B showed it matches with some theories, is it possible to provide some data to show the results are converged with sufficient sampling (like longer simulations) for both single spectrin and spectrin networks? A brief description of the difference between Flory theory and WLC/FJC models will be helpful.

6. The lipid bilayer membrane may play an important role, but it is assumed to a flat plane. Will the flexibility and fluctuations of the lipid bilayer impact the process simulated?

7. In addition, using harmonic constraint to fix all particles, including spectrin and actins in 4nm thick plane 10 nm above the bilayer seems very artificially. What are the physical justifications? If they are confined in 4 nm plane 10 nm above, how these thickness of the cytoskeleton (54.21 nm) are estimated later on? Did I misunderstand something?

8. If keeping decreasing the shear rate, will the shear modulus curve converge? The shear rate effect might be artificial, since the hydrodynamic effect is not captured accurately in Brownian dynamics. In addition, what are the major sources of dissipation if it is rate-dependent? Is it from spectrin? But spectrin does not have any dissipation term in the current model. Maybe from the lipid bilayer, but it is not modeled.

9. In Figure 4, is it possible to plot shear modulus as a function of shear?

10. It is good to compare the shear responses between simplified model and the full model in Figure 4. But is it possible to plot the case with actin polymerization versus the case without it? Will actin polymerization change the behavior?

11. Considering the binding between actin, spectrin, and KAHRP can be very helpful for understanding the mechanisms, but it is known the spectrin tetramer can dissociate into dimers? Will modeling this dissociation change the picture? The rate constants were published before for dime association.

12. How exactly does the spectrin bind to the polymerized actin filament is not clearly described? It is known that spectrin bind to specific locations of the actin filament. How is this achieved in the algorithm? Will the spectrin only bind to certain bead in the actin? Do we know the binding constants between actin and beta spectrin CH end? How to prevent multiple spectrins bind to the same site of the actin if LJ potential is used? Figure 2CD show the LJ potential between actin and spectrin has an equilibrium length of 8nm, any physical justification? Spectrin-spectrin bond is 12 nm in Fig. 2C, but one spectrin repeat is only 5.2 nm as shown in Fig. 2A. Is this a mistake?

13. What are the grey particles in the simulations? Are they G actins? Will G actin be able to nucleate by itself? Any treadmilling behavior?

14. The actin polymerization modeling might be related to the work by Fowler group, but Fowler group also believe the myosin plays a major role in red cell mechanics. Will it be difficult to consider myosin in the current model or any discussion on why it is not important in the current study?

15. It’s claimed that the interaction between KAHRP and ankyrin/actin are more important than its interaction with spectrin. Any data to support this claim?

16. 140 x242:48 x100 nm seems too small comparing to existing similar studies. A representative volume element needs to include sufficient network to get reasonable spatial averaging, unless the authors can show when they make the patch larger it won’t impact the result. The authors used 9 repeats patch later on. Are these small and big patches giving the same results for stress and shear modulus?

17. Figure 2D shows the initial configuration of the network. Are all spectrins with the same initial length ‘a’ in Fig. 2D? It seems unlikely due to the PBC condition and random orientation of actins. How are the spectrin initial length determined if they are not the same? Fig. 2D also shows the actin consistent of 6 beads, but the actual actin filament has about 13 monomers. If one beads represents 2 monomers, how the rate of polymerization changes, since I believe the rates are obtained between monomers?

18. How high is the temperature in these simulations? Does the change in temperature has any effect on results or simulations process, room vs body?

19. There should be bonded connection between actins and the lipid bilayer through glycophorin C, p55, protein 4.1 et al. How will this impact the actin dynamics? Will it be possible to include in the model?

20. Finally, the study of the effect of KAHRP on mechanical properties is interesting, but what are the physical mechanisms? After KAHRP binding to the actin or ankyrin, will it prevent other proteins, such as spectrin, to bind? Or will it weaken or strengthen these existing binding? Or it does not change these, but the mass or diffusivity? In these discussions, the authors talked about ‘increased initial step’ or ‘increased jump’, are they step or jump of stress? At this scale, viscous force is much larger than inertial force. Does this mean the cluster and the larger junction has a larger diffusivity, which lead to altered stress? In summary, it is unclear what are the major changes in terms of components and interactions between them caused by KAHRP.

Reviewer #3: In this paper, the authors present a detailed model for the RBC cytoskeleton using Brownian Dynamics (using a software package called ReADDY). They focus on the binding of KHARP, a protein that is associated with the malaria parasite, Plasmodium falciparum. They go on to describe a detailed model of the RBC cytoskeleton and the extract its shear modulus under different conditions. The work is thorough and detailed. Since the code was not publicly shared, I cannot comment on whether the results are reproducible. Given the results, I have the following comments

1. How is the membrane represented in this system? Given that the RBC cytoskeleton is closely associated with the membrane, is the membrane even represented in this system? The knobs of KHARP are transmembrane as described in Figure 2. Without any discussion of this, it is hard to evaluate how the results make sense. At the very minimum, the authors should discuss this omission and how it may impact their results.

2. There are quantitative measurements of RBC cytoskeleton arrangements in two papers the authors cite (ref 17 and 18). Also, ref 17 and 18 contradict each other. So how do the authors results compare against these references? A quantitative comparison is possible with the data the authors have but has not been included. I believe it is a missed opportunity.

3. Nonmuscle myosinIIA (NMIIA) is conspicuously missing from the model. See https://www.pnas.org/content/115/19/E4377.short. The authors should comment on the role played by NMIIA potentially.

4. Finally there is only one result with respect to malaria in Figure 6 and even there it is not clear what the different lines in Figures E and F are (legend has only one value). But it is clear that these trends don't match very well with Figure 6B and C and further explanation is needed as to why that may be.

**Have the authors made all data and (if applicable) computational code underlying the findings in their manuscript fully available?**

Reviewer #1: Yes

Reviewer #2: Yes

Reviewer #3: **No: **The submission says Code will be available upon reasonable request.

PLOS authors have the option to publish the peer review history of their article (what does this mean?). If published, this will include your full peer review and any attached files.

Reviewer #1: No

Reviewer #2: No

Reviewer #3: No
---

## [Decision Letter · Decision Letter 1]

21 Mar 2022

Dear Prof. Dr. Schwarz,

We are pleased to inform you that your manuscript 'A particle-based computational model to analyse remodelling of the red blood cell cytoskeleton during malaria infections' has been provisionally accepted for publication in PLOS Computational Biology.

Best regards,

Alison L. Marsden

Associate Editor

PLOS Computational Biology

Jason Haugh

Deputy Editor

PLOS Computational Biology

Reviewer's Responses to Questions

**Comments to the Authors:**

Reviewer #1: The authors have comprehensively revised and answered all my previous comments.

Reviewer #2: none.

Reviewer #3: The authors have addressed all the comments I raised in the previous version and I'm satisfied with the changes made to the manuscript.

**Have the authors made all data and (if applicable) computational code underlying the findings in their manuscript fully available?**

Reviewer #1: Yes

Reviewer #2: None

Reviewer #3: Yes

PLOS authors have the option to publish the peer review history of their article (what does this mean?). If published, this will include your full peer review and any attached files.

Reviewer #1: No

Reviewer #2: No

Reviewer #3: No

---

## [Editor Report · Acceptance letter]

1 Apr 2022

PCOMPBIOL-D-21-01776R1 

A particle-based computational model to analyse remodelling of the red blood cell cytoskeleton during malaria infections

Dear Dr Schwarz,

I am pleased to inform you that your manuscript has been formally accepted for publication in PLOS Computational Biology. Your manuscript is now with our production department and you will be notified of the publication date in due course.

With kind regards,

Zsanett Szabo
